# Comparisons of Estimated Intakes and Plasma Concentrations of Selected Fatty Acids in Pregnancy

**DOI:** 10.3390/nu11030568

**Published:** 2019-03-06

**Authors:** Marie T. B. Madsen, Anne A. Bjerregaard, Jeremy D. Furtado, Thorhallur I. Halldorsson, Marin Ström, Charlotta Granström, Edward Giovannucci, Sjurdur F. Olsen

**Affiliations:** 1Department of Epidemiology Research, Statens Serum Institut, DK-2300 Copenhagen, Denmark; ANNE@ssi.dk (A.A.B.); LUR@ssi.dk (T.I.H.); MRM@ssi.dk (M.S.); CGS@ssi.dk (C.G.); sfo@ssi.dk (S.F.O.); 2Department of Nutrition, Department of Epidemiology, Harvard T.H. Chan School of Public Health, Boston, MA 02115, USA; jfurtado@hsph.harvard.edu (J.D.F.); egiovann@hsph.harvard.edu (E.G.); 3Faculty of Food Science and Nutrition, School of Health Sciences, University of Iceland, 101 Reykjavik, Iceland; 4Faculty of Natural and Health Sciences, University of Faroe Islands, Torshavn 100, Faroe Islands

**Keywords:** long-chained n-3 fatty acids, LCn3FAs, polyunsaturated fatty acid, PUFAs, food frequency questionnaire, FFQ, validation study, biomarkers, Danish National Birth Cohort, pregnancy diet

## Abstract

The growing interest in potential health effects of long-chain polyunsaturated fatty acids (PUFAs) makes it important to evaluate the method used to assess the fatty acid intake in nutrition research studies. We aimed to validate the questionnaire-based dietary intake of selected PUFAs: eicosapentaenoic acid (EPA), docosahexaenoic acid (DHA), α-linolenic acid (ALA), linoleic acid (LA), and arachidonic acid (AA) within the Danish National Birth Cohort (DNBC), by comparing 345 women’s reported intake with concentration of plasma biomarkers. The applied questionnaire- and biomarker data reflect dietary intake from around the same time point in mid-pregnancy and relationships were investigated by use of Pearson and Spearman correlation and linear regression statistics. We demonstrated moderate but consistent adjusted correlations between dietary intake estimates and the corresponding plasma biomarker concentrations (differences in plasma concentration per 100 mg/day greater intake of 0.05 (95% CI: 0.02; 0.08)) and 0.05 (95% CI: 0.01; 0.08) percentage of total plasma fatty acids for EPA and DHA, respectively). The associations strengthened when restricting the analyses to women with ALA intake below the median intake. We found a weak correlation between the dietary intake of ALA and its plasma biomarker with a difference in plasma concentration of 0.07 (95% CI: 0.03; 0.10) percent of total plasma fatty acids per 1 g/day greater intake, while the dietary intake of LA and AA did not correlate with their corresponding biomarkers.

## 1. Introduction

Polyunsaturated fatty acids (PUFAs) are commonly perceived as health-promoting dietary components [1]. Particularly, the seafood-specific PUFAs, eicosapentaenoic acid (EPA), and docosahexaenoic acid (DHA), belonging to the essential long-chained *n*-3 (LCn3FAs) family, have been investigated thoroughly in the attempt to understand their putative beneficial roles in pregnancy-related complications, developmental outcomes, as well as non-communicable diseases in adulthood [2,3,4,5,6]. This particular interest in dietary PUFAs makes it important to evaluate the various methods used in nutrition research to estimate fatty acid intake, since a poor estimate might lead to distorted associations between health and disease endpoints, and consequently inadequate dietary recommendations based on the false or imprecise findings.

Self-reported dietary intake is a commonly employed measure in epidemiological studies, and has a high utility, such as in relation to establishing dietary recommendations. However, estimates based on self-reporting are often afflicted by lack of precision in the reporting and are curtailed by the methods used for nutrient calculations, which often have to rely on quite rough assumptions related to standard recipes, portion sizes, and food compositions [7].

Plasma EPA and DHA concentrations are already recognized as specific and robust biomarkers of EPA and DHA status and intake as indicated by strong correlations with dietary intake estimates [8,9]. Other commonly addressed PUFAs in diet–disease association studies include α-linolenic acid (ALA), linoleic acid (LA), and arachidonic acid (AA). Some studies show moderate to strong correlations between intakes of LA and ALA and their plasma biomarkers [10,11,12]; however, overall, plasma levels of ALA, LA, and AA tend to show much weaker correlations with estimated dietary intake compared with EPA and DHA [11,13,14,15]. Whether the poor correlations are caused by imprecision in intake quantification or inconsistency between actual dietary intake and plasma levels due to endogenous processes is not completely understood.

The food frequency questionnaire (FFQ) applied in the large Danish National Birth Cohort (DNBC) has previously been validated in a subsample of 88 women against 7-day weighted food diaries and blood biomarkers of LCn3FAs [16]. The results demonstrated a moderate correlation between FFQ-based intake estimates of total LCn3FAs and EPA levels in erythrocyte membranes (*r* = 0.37, *p* = 0.001) [16]. In the present study, we had an opportunity to explore the relationship in a bigger subsample of the DNBC (345 vs. 88 women in the earlier study), for a wider range of fatty acids, and with a more optimal design for studying the relationship between self-reported dietary intake and biomarkers.

To justify the use of FFQ-derived estimates of long chained PUFA intake in the Danish National Birth Cohort (DNBC) in future diet–disease association studies, the aim of the present study was to validate the FFQ-based intake estimates of EPA and DHA and investigate—in a more exploratory manner—the relationship between ALA, LA, and AA intake and their respective biomarkers.

## 2. Materials and Methods

### 2.1. The Cohort Database

The DNBC was established in the period of 1996–2002 and included approximately 100,000 pregnancies in Denmark. All participating women gave consent on behalf of themselves and their children to participate in interviews and questionnaires, giving blood samples, register linkages, and other types of follow-up (Figure 1) [17]. Blood from the DNBC women was collected around gestation week (GW) 10 and 25 as well as from the umbilical cord at birth (referred to as blood sample (BS) 1, 2, and 3, respectively) [17]. 

### 2.2. Dietary Fatty Acid Intake

At the time of signing informed consent (GW 6–10), the women also answered questions about their use of drugs and dietary supplements during the peri-conceptional period, providing information on up to eight different drugs and supplements. Telephone interviews (TI) were scheduled to be undertaken in GW 12 and GW 30. The women completed a comprehensive FFQ with approximately 360 food items in GW 25–26, which captured foods and beverages consumed in the past four weeks [18]. Central food items in the LCn3FAs intake calculation include seafood cold cuts (12 types) and warm meals with seafood (14 types). Some important contributors to ALA, LA, and AA were plant oils and margarines for cooking and as spread, dressings, eggs, meats, and processed foods. The most commonly used plant oils used for cooking in the DNBC included sunflower, olive, and grapeseed oil. Intake in grams or milligrams per day for all fatty acids was computed based on the FFQ-reported daily intake frequencies of food items multiplied by predefined portion sizes, followed by nutrient calculations by use of Danish nutrient tables developed and continuously updated by the National Food Institute at the Technical University of Denmark [19]. In addition to dietary intake, the FFQ collected information on dietary supplementation by open-ended questions, which allowed an indication of adherence of and details on up to six different dietary supplements taken during pregnancy, including product name, nutrient content, and dosage.

### 2.3. Plasma Fatty Acid Concentration

Comprehensive plasma fatty acid concentration analyses of EPA, DHA, ALA, LA, and AA from BS1 and BS2 were conducted at the Nutritional Biomarker laboratory at Harvard T. H. Chan School of Public Health (HSPH) from January to November 2015. The samples for the present study were frozen and stored at −30 °C at first, and in 2013 allocated to either −20 (3.6%), −80 (15.9%), or −196 °C (82.6% of samples) freezers in the Danish National Biobank [17]. Primarily, we aimed to compare FFQ-data with BS2, since it was drawn at mid pregnancy where the FFQ was filled out. If BS2 was missing and BS1 from GW 10 was available, BS1 was used instead (1.5%). Fatty acids were extracted and prepared for analysis by transmethylation, using methanol and sulfuric acid as described previously [20,21]. Following esterification, the fatty acid methyl esters were suspended in iso-octane and quantitated by gas-liquid chromatography as fully described by Baylin et al. [10]. The fatty acid measurements represented the complete spectrum of plasma lipid fractions, including triacylglycerol, cholesterol esters, phospholipids, and free fatty acids (including albumin-bound free fatty acids). Quality control in the HSPH laboratory is maintained by external validation through participation in programs offered by both the American Oil Chemists Society and the National Institute of Standards and Technology. EPA, DHA, ALA, LA, and AA have within-run coefficients of variation of 2%, 2%, 1%, 1%, and 2%, respectively. The between-run coefficients of variation for the fatty acids are 7%, 11%, 4%, 2%, and 6%, respectively.

A randomly selected control group of 345 women analyzed in relation to a previous case-control study nested in DNBC [22]. We excluded women with missing FFQ-data (*n* = 79), women reporting the use of fish oil supplements (*n* = 10), and one woman with an unrealistic linoleic acid intake (78 g/day) (Figure 2).

### 2.4. Statistical Analyses

Data was analyzed with the SAS software package (version 9.4; SAS Institute Inc., Cary, NC, USA) and the statistical analysis plan was defiend *a priori*. We used 0.05 as the cut-off value for statistical significance. Preliminary testing for linearity and homoscedasticity was done using scatterplots and testing the variable assumption of normal distribution was done using histograms and QQ-plots. Population characteristics were evaluated by univariate descriptive statistics. 

EPA and DHA data was analyzed individually as well as an aggregated EPA and DHA variable, from this point referred to as “EPA + DHA”. ALA, LA, and AA were analyzed individually. In the primary analysis, the dietary intake of fatty acids was expressed as absolute daily intake/day. Plasma fatty acid concentrations were expressed as a percentage of total plasma fatty acids (%TPFA) in plasma. We conducted multivariable linear regression analyses to examine the association between dietary intake and the plasma biomarkers of the fatty acids. The intake variable was computed as explanatory and the biomarker variable as the response variable in these models. EPA and DHA are statistically evaluated and presented as 100 mg/day, ALA and LA as g/day, and AA as 10 mg/day to obtain intuitively interpretable results according to typical daily intakes. In addition, an adjusted model was computed to account for potential interfering covariates identified *a priori* from the literature [8]. Covariates included age, body mass index (BMI), parity, smoking, and total energy intake. Some women were excluded (total of 5.5%) from the adjusted analyses due to missing data (for age, BMI, parity and smoking 0.4%, 5.1%, 4.3% and 0.4%, respectively). In order to compare with previous studies, we performed correlation analyses, as this is the most frequently reported estimate, from similar published studies. Depending on the nature of the variables, the non-parametric test Spearman correlation (for EPA, DHA, and LA) or the parametric test Pearson correlation (for ALA and AA) statistics was applied. 

### 2.5. Secondary Analyses

As ALA was expected to contribute to some of the plasma content of EPA and DHA due to endogenous conversion, we stratified the women by intakes higher and lower than the cohort’s median in FFQ-based ALA intake. In the supplementary analyses, we applied the relative intake of specific fatty acids (percentage of total intake of fatty acids) instead of absolute intake of fatty acids (g/day) to investigate the effect of total fat intake. Moreover, we evaluated the implication of higher storage temperatures than −196 °C during the last 2 years by excluding samples stored at −20 °C and −80 °C and samples with missing temperature.

The study was conducted in accordance with the Declaration of Helsinki, and all study procedures were approved by the Regional Scientific Ethics Committee for the municipalities of Copenhagen on 27 August 2013 (H-2-2013-108).

## 3. Results

Most women were primiparous, non-smokers, had high school or higher level of education, and consumed a typical Danish diet in terms of energy intake and macronutrients composition [23] (Table 1).

Median intakes of EPA and DHA were estimated to 0.10 and 0.26 g/day, respectively. Median intakes of ALA, LA, and AA were 2.06, 9.81, and 0.08 g/day, respectively. Plasma concentrations of EPA and DHA were 0.45 and 1.53%TFPA, respectively. For ALA, LA, and AA the plasma concentrations were 0.58, 19.86, and 3.08%TFPA, respectively (Table 1).

Linear regression of the biomarker concentration on estimated intake resulted in adjusted differences in plasma concentration of 0.05 (95% CI: 0.02; 0.08) and 0.05 (95% CI: 0.01; 0.08) %TPFA per 100 mg/day greater intake of EPA and DHA, respectively (Table 2). The unadjusted relationship is illustrated in Appendix A.

Estimated ALA intake showed a weak association with the plasma biomarker that was pronouncedly strengthened when adjusting for covariates in the linear regression model. This resulted in a difference in ALA concentration of 0.07 (0.03; 0.10) %TPFA per 1 g/day greater ALA intake. ALA intake did not correlate with EPA+DHA plasma concentrations (*p* = 0.42, Spearman correlation). Estimated intakes of LA and AA did not correlate with their corresponding plasma biomarkers (Table 2).

When stratifying according to estimated intake of ALA (below or above the median intake of 2.06 g/day) (Table 3), women with low ALA intakes tended to have stronger correlations between intakes and plasma biomarkers of EPA and DHA compared to the women with high intakes of ALA. (Table 3).

When expressing the intake of specific fatty acids as percentage of total fatty acid intake (Appendix A) instead of absolute intake (Table 2), the correlation coefficient increased for DHA, which is also demonstrated by lower *p*-value in the unadjusted linear regression coefficient. For ALA, this weakened the strength of correlation to less than significance level. LA and AA, expressed as percentage of total fatty acid intake, remained insignificant in their association with biomarkers (Appendix A).

Both EPA and DHA correlations were somewhat weakened when we restricted the analysis to samples stored at −196 °C, thus excluding samples stored at −20 °C and −80 °C and missing temperatures (*n* = 52 excluded), but the overall associations tended to be retained for these fatty acids. This stratification did not change the correlations for ALA, LA, and AA (Appendix A).

## 4. Discussion

In this study, we demonstrated moderate but consistent correlations between FFQ-based dietary intake estimates of EPA and DHA and the corresponding plasma biomarker concentrations. The associations tended to get stronger when restricting the analysis to women with low intake of ALA. We found a weak correlation between the dietary intake of ALA and its plasma biomarker, while dietary intake of LA and AA failed to correlate with the corresponding plasma biomarkers. 

EPA and DHA—Our results are generally in accordance with previous findings in non-pregnant, addressing the relationship between self-reported intake of EPA and DHA from any type of assessment method: FFQs, food diaries or records, interviews, and biomarkers of EPA and DHA from various body tissues [8,9]. Published correlation coefficients applying the same combination of data sources (FFQ and biomarkers in plasma) range from *r* = 0.23 in a Costa Rican population (*n* = 200) [10] to *r* = 0.54 in an Australian population (*n* = 53) for total n-3 fatty acids [24]. Within these extremes, many intermediate estimates have been reported [11,12,13,14,15,24,25]. In addition to the varying population sizes, the discrepancy in correlation strength possibly arises from differences in FFQ quality as well as in lifestyle, that is, ranges of intake and genetic as well as metabolic characteristics between the populations. Additionally, the choice of biomarker tissue, lipid fractions, sample processing, fatty acid separation, and quantification technique might also contribute to the between-study differences.

Conceptually, it is reasonable to believe that metabolic changes in pregnancy can disrupt the normal relationship between intake and plasma levels of fatty acids, which challenges comparison of present results with previous validation studies in non-pregnant populations. As an example, circulating estrogen is increased during pregnancy, and estrogen is known to stimulate DHA synthesis [26]. Furthermore, in contrast to adipose tissue, plasma is affected by metabolic variation between individuals such as the levels of HDL, LDL, and VLDL in plasma, since they contain different proportions of the fatty fractions: triglycerides, cholesterol esters, and phospholipids which in turn will have distinct fatty acid compositions [8]. During pregnancy, increases in LDL and VLDL are commonly observed [27], which could possibly differentiate present results from other validation studies based on non-pregnancy cohorts. Furthermore, the challenge of reporting error may be more pronounced in pregnant women as dietary habits are expected to be more dynamic due to pregnancy-related behavioral changes [28].

Dietary habits including total fat intake might play a role in the diet–biomarker relationship [8]. In the present study, total fat intake was indirectly accounted for by expressing the intake variable as percentage of total fat intake and this maintained a correlation between LCn3FAs intakes and their biomarkers with strengthened correlation for DHA. Another important source to between-study variation is differences in between-person variation in LCn3FA intake across studies, which, if large, gives rise to stronger diet-biomarker association correlation coefficients [10,14]. This phenomenon is reflected in results from a Japanese cohort, reporting a median consumption of around four times the EPA and DHA intake in DNBC [29]. The higher intake among Japanese participants gives rise to greater variability in absolute intake than for DNBC, which might have contributed to the strong intake–biomarker correlations of *r* = 0.43 and *r* = 0.35 for EPA and DHA, respectively [29]. The intake estimate of 0.37 g EPA+DHA per day among DNBC women is low compared to previous estimations within the Danish population. Intake estimates of EPA+DHA deriving from 7-day weighted food records from women (*n* = 63) and 7-day food record from women and men (*n* = 24) have been determined to approximately 0.5 g/day in both studies [30,31].

High or low consumption of other fatty acids from the n-3 and n-6 families may have affected the investigated correlations [32]. The process of transforming n-3 fatty acids into the long-chained fatty acids EPA and DHA is highly dependent on the accessibility of ALA, since it serves as substrate for endogenous LCn3FAs synthesis. Therefore, a low ALA intake is expected to limit the formation of EPA and DHA and vice versa. In line with the theory, we observed that the women with a low intake of ALA had a stronger correlation for both EPA and DHA compared to women with a high intake. Interestingly, and as also reported by others, EPA correlated better with its corresponding biomarker than DHA, which might be explained by binding affinity differences between the two fatty acids in the chylomicron-dependent transport to the liver following absorption [32].

Plasma lipids may have been affected by several non-dietary factors, some of which we were able to adjust for in the regression models. In addition to the covariates included in the adjusted model (age, parity, BMI, smoking, and total energy intake), we know from previous literature that other modifiable and likely pregnancy-dependent factors including sex hormones and exercise can influence plasma levels of LCn3FAs [8]. Genetic differences might also contribute to the between-study differences in diet–biomarker correlations. Genes coding for proteins involved in uptake, metabolism, transport and restructuration, especially the FADS gene cluster, which is related to desaturation steps in the n-3 and n-6 fatty acids, could potentially differentiate the DNBC women from other study populations [8].

**ALA, LA, AA**—While a correlation between FFQ-reported intake and plasma biomarkers of EPA and DHA is well established, it remains more speculative for the remaining members of the PUFA family. It seems plausible that higher availability of the n-6 family fatty acids in plasma compared to n-3 family fatty acids makes endogenous conversion, restructuring to other fat fractions or spillover to other body compartments more pronounced for the n-6 family [8]. Such endogenous dynamics of ALA, LA, and AA challenge the use of biomarkers to reflect dietary intake and may have played an important role in the weak correlations observed. Previous results, however, demonstrate a marked variation in intake and biomarker correlation for LA and generally low to moderate correlation for ALA and poor correlation for AA [8].

Limitations in the self-reported dietary intake, technical errors in the plasma measurements, or actual between-individual differences in the study population might play a role in the lack of correlation between estimated dietary intake and biomarker level. It is most likely a function of all three, but what clearly differentiates the results on ALA, LA and AA from those on EPA and DHA is the greater difficulty in dietary intake assessment. EPA and DHA are found in high concentrations in a limited number of foods of marine origin, and which therefore can be assessed by a FFQ with relatively few food items. The long list of dietary sources of ALA, LA hereunder the cooking oils (sunflower, olive and grapeseed oil) used by DNBC women and the various animal products containing AA make assessment accuracy highly dependent on the quality and coverage of the FFQ and the reporting performance. For these foods, the nutritional composition, serving size, and frequency of intake are difficult to determine by a FFQ.

Plasma LA levels has previously been reported to be a valid biomarker of dietary LA intake in the Costa Rican study (*r* = 0.41) [10]. In the Costa Rican cohort, much effort has been allocated to biochemical assessment of the fatty acid composition of the actual oil product consumed, which may have yielded a higher accuracy of the reported dietary intake of LA compared to our data. The DNBC FFQ captured the intake of plant oils by addressing oils used for cooking and frying and in dressings/spreads, but not as comprehensively as in the Costa Rican cohort study. This could indicate that the FFQ’s inaccuracy in capturing LA intake could be the driver of the weak LA correlation observed.

There is an ongoing debate about which biological tissue provides the fatty acid biomarkers that best represent dietary intake. Adipose tissue is commonly used and generally considered the best tissue to reflect long-term dietary intake [9] but might not be appropriate to reflect more recent intake. Previous evaluation found that LCn3FA levels in plasma correlate well with adipose tissue levels [10]. Furthermore, albeit with weaker correlation than for fatty acids in adipose tissue, plasma fatty acids have previously been demonstrated to correlate well with FFQ-derived dietary intake [10]. We analyzed total lipid fractions in plasma, which include phospholipids, triglycerides, cholesterol esters, and free fatty acids. Since some fractions reflect long-term dietary intake while others reflect short-term fluctuations in the diet, it will limit the temporal specificity of the measurement, and might not reflect same time window as the FFQ. Free fatty acids in plasma have previously been shown to correlate well with the long-term fatty acid biomarkers from subcutaneous adipose tissue, while phospholipids and cholesterol esters reflect the previous week’s intake and triglycerides reflect the fatty acid composition of the latest meal [8]. However, other experimental studies show that there are no differences between triglycerides, cholesterol esters, and phospholipids in response to short-term increases in fatty acid intake [32,33], indicating that it is still a subject of controversy. Another concern in present study is the non-fasting state of the women when samples were drawn. However, several studies show minimal effect of fasting, and a recent evaluation from European Federation of Clinical Chemistry and Laboratory Medicine suggests non-fasting samples as the new standard medium for lipid measurement [34].

## 5. Conclusions

In summary, our findings suggest that the DNBC FFQ adequately quantifies DHA and EPA intakes in Danish pregnant women, supporting their use in future diet–disease association studies within DNBC. We found a weak correlation between ALA intake estimate and its biomarkers, and no correlation for LA and AA. Whether this is caused by poor validity of the intake estimates or by biologically-based variability between intake and plasma levels is yet to be investigated. Noticeably, our results are specific for pregnant women, who might differ from other non-pregnant study populations.

## Figures and Tables

**Figure 1 nutrients-11-00568-f001:**
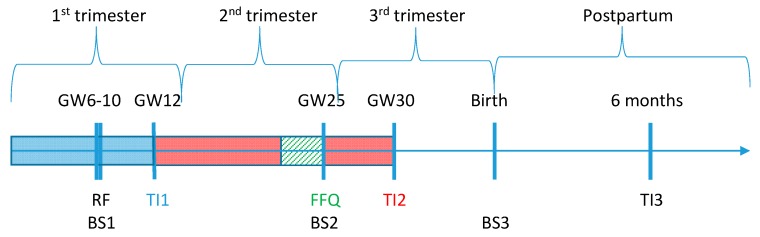
Timeline for data collection in Danish National Birth Cohort (DNBC). Colored boxes show the retrospective recall period for reported dietary intake. BS: blood sample; FFQ: food frequency questionnaire; GW: gestational week; RF: recruitment form; TI: telephone interview.

**Figure 2 nutrients-11-00568-f002:**
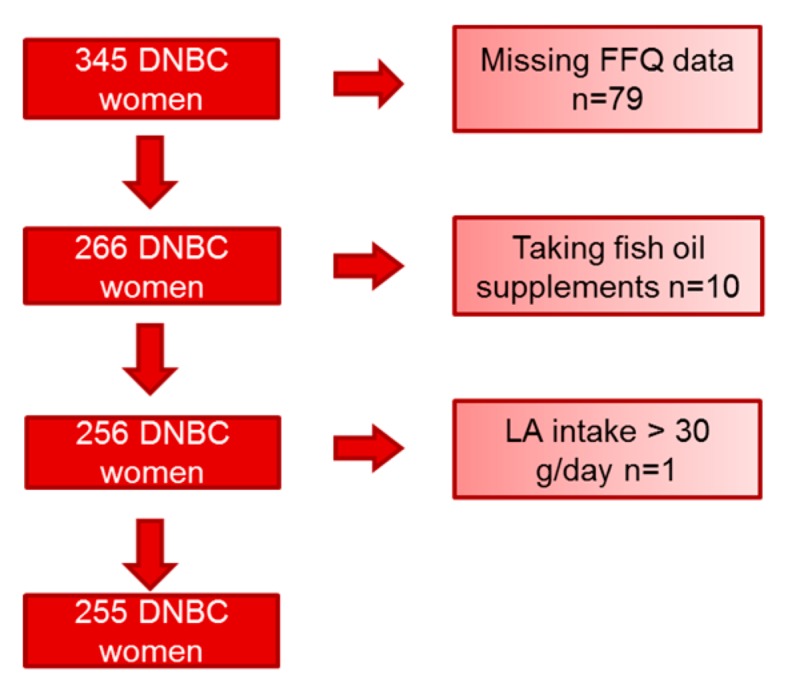
Participant exclusion chart. DNBC; Danish National Birth Cohort; LA: linoleic acid.

**Table 1 nutrients-11-00568-t001:** Characteristics of participants at enrollment and dietary intake at mid pregnancy (*n* = 255).

**Age, years**	29.5 (4.4)
**BMI, kg/m^2^**	23.2 (4.2)
**Parity, n (%)**	
0 children	118, 51%
1 child	84, 36%
≥2 children	30, 13%
**Smoking, n (%)**	
Non-smoker	177, 70%
Occasional smoker	26, 10%
Daily smoker <15 cigarettes/day	43, 17%
Daily smoker ≥15 cigarettes/day	8, 3%
**Education, n (%) 201**	
10th grade or less	53, 26%
High school similar or more	140, 70%
**Dietary intake**	
Total energy intake, kJ/day	10,210 (2550)
Carbohydrate, % energy	51.8 (5.9)
Fat, % energy	32.6 (6.3)
Protein, % energy	15.2 (2.5)
Alcohol, % energy	0.4 (0.5)
EPA, g/day	0.10 (0.12) *
DHA, g/day	0.26 (0.25) *
EPA + DHA, g/day	0.37 (0.35) *
ALA, g/day	2.06 (1.17) *
LA, g/day	9.81 (4.36) *
AA, g/day	0.08 (0.06) *
**Plasma biomarker (%TPFA)**	
EPA	0.45 (0.21)
DHA	1.53 (0.55)
EPA + DHA	1.99 (0.73)
ALA	0.58 (0.15)
LA	19.86 (3.12)
AA	3.08 (1.02)

Presented as number, % for categorical variable and mean (sd) for continuous variables unless otherwise indicated. A random sample of pregnancies from the Danish National Birth Cohort (*n* = 243). * =Median (IQR)AA: arachidonic acid; ALA: α-linolenic acid; BMI: body mass index; DHA: docohexaenoic acid; EPA: eicosapentaenoic acid; IQR: inter quartile range; LA: linoleic acid; %TPFA: percentage of total plasma fatty acids.

**Table 2 nutrients-11-00568-t002:** Difference in biomarker concentration (%TPFA) per increase in daily intake and correlations between biomarker concentration and intake from Spearman statistics unless indicated otherwise (n = 255).

	**Regression Coefficients (95% CI) from Linear Regressions**		**Correlation Coefficients**
	**Non-Adjusted**	**Adjusted for Age, Parity, BMI, Smoking, and Total Energy Intake**	
**EPA**	**0.05 (0.02; 0.08)** % per 100 mg/day ******	**0.05 (0.02; 0.08)** % per 100 mg/day *****	**0.24 *****
**DHA**	**0.05 (0.01; 0.08)** % per 100 mg/day *****	**0.05 (0.01; 0.08)** % per 100 mg/day *****	**0.18 ***
**EPA + DHA**	**0.05 (0.02; 0.08)** % per 100 mg/day *****	**0.05 (0.02; 0.08)** % per 100 mg/day *****	**0.21 ****
**ALA**	**0.03 (0.01; 0.05)** % per 1 g/day	**0.07 (0.03; 0.10)** % per 1 g /day ******	**0.15 ^a^**
**LA**	0.10 (−0.01; 0.22) % per 1 g/day	0.17 (−0.04; 0.38) % per 1 g/day	0.08
**AA**	0.01 (−0.02; 0.03) % per 10 mg/day	0.02 (−0.01; 0.04) % per 10 mg/day	0.05 **^a^**

**BOLD** = *p* < 0.05, * = *p* < 0.01, ** = *p* < 0.001, *** = *p* < 0.0001. ^a^ Pearson correlation. AA: arachidonic acid; ALA: α-linolenic acid; BMI: body mass index; DHA: docohexaenoic acid; EPA: eicosapentaenoic acid; LA: linoleic acid; %TPFA: percentage of total plasma fatty acids.

**Table 3 nutrients-11-00568-t003:** Difference in biomarker concentration (%TPFA) per increase in daily intake and correlations between biomarker concentration and intake from Spearman statistics unless indicated otherwise.

		**Regression Coefficients (95% CI) from Linear Regressions**		**Correlation Coefficient**
		**Non-Adjusted**	**Adjusted for Age, Parity, BMI, Smoking and Total Energy Intake**	
**EPA**	≥2.06 g/day ALA	**0.05 (0.01; 0.09)** % per 100 mg/day	**0.05 (0.01; 0.09)** % per 100 mg/day	0.18
	<2.06 g/day ALA	**0.06 (0.02; 0.11)** % per 100 mg/day *	**0.05 (0.01; 0.10)** % per 100 mg/day *	**0.33 ****
**DHA**	≥2.06 g/day ALA	0.03 (−0.02; 0.07) % per 100 mg/day	0.02 (−0.02; 0.07) % per 100 mg/day	0.09
	<2.06 g/day ALA	**0.09 (0.04; 0.15)** % per 100 mg/day *	**0.09 (0.03; 0.14)** % per 100 mg/day *	**0.27 ***
**EPA + DHA**	≥2.06 g/day ALA	**0.03 (−0.01; 0.07)** % per 100 mg/day	**0.03 (−0.01; 0.07)** % per 100 mg/day	0.12
	<2.06 g/day ALA	**0.08 (0.03; 0.14)** % per 100 mg/day *	**0.08 (0.03; 0.13)** % per 100 mg/day *	**0.29 ****

Stratified analyses according to dietary ALA-above (*n* = 128) and below (*n* = 127) median intake of ALA (2.06 g/day). **BOLD** = *p* < 0.05, * = *p* < 0.01, ** = *p* < 0.001. AA: arachidonic acid; ALA: α-linolenic acid; DHA: docohexaenoic acid; EPA: eicosapentaenoic acid; LA: linoleic acid; %TPFA: percentage of total plasma fatty acids.

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
