# Peer review of "Comparisons of Estimated Intakes and Plasma Concentrations of Selected Fatty Acids in Pregnancy"

_nutrients, 2019, doi:10.3390/nu11030568_

Round 1

Reviewer 1 Report

In general, this manuscript is well-written. Analyses among pregnant women provides some novelty for the study findings.

Minor comments:

Methods:

Coefficients of variation (CV%) should be shortly presented for different plasma fatty acid biomarkers (page 3, line 109).

Discussion:

Free fatty acids are not located in lipoproteins in plasma (page 7, line 226). They are mainly albumin-bound non-esterified fatty acids.

"We analyzed total lipid fractions in plasma......" (page 8, line 298). Similarly, free fatty acids are mainly albumin-bound fatty acids.

Author Response

Thank you very much for your good and relevant comments!

1.     Coefficients of variation (CV%) should be shortly presented for different plasma fatty acid biomarkers (page 3, line 109).

Yes indeed, it has been added to the method (in section “Plasma fatty acid concentration”)

2.       Free fatty acids are not located in lipoproteins in plasma (page 7, line 226). They are mainly albumin-bound non-esterified fatty acids.

Agree and corrected (see attached manuscript)

3.     "We analyzed total lipid fractions in plasma......" (page 8, line 298). Similarly, free fatty acids are mainly albumin-bound fatty acids. 

We do consider FFA (free fatty acid) to be comprised of both albumin-bound and truly “free” fatty acid.  There is a very tiny fraction of truly “free fatty acid”, but the terms “free fatty acid” and “non-esterified fatty acid” are generally used interchangeably and denote any fatty acid not bound to cholesterol, a phosphate group, a glycerol molecule, etc. This is now explained in the method section – thank you for pointing out the unclearness.

Have a good day, 

Marie 

Reviewer 2 Report

Comparisons of estimated intakes and plasma concentrations of selected fatty acids in pregnancy

Madsen el al. examined relationships between dietary intakes and circulating concentrations of 5 common PUFA in pregnant women participating in the Danish National Birth Cohort. This study expanded a previous report in the same cohort by adding more individuals (n=255, following exclusions) and more fatty acids (AA n-6, ALA n-3, DHA n-3, EPA n-3, LA n-6). The focus on pregnant women, whose metabolism is altered to accommodate fetal development, is of interest.  Comments are generally intended to clarify the presentation of results and overall meaning, especially when the results refer to dietary FA and when to circulating (plasma) FA.

Specific comments

1.     The Section on secondary analyses mentions 3 storage temperatures, -20 C, -80 C and -196 C. Please indicate storage temperature in the methods for “Plasma FA concentration” section, where readers will look for it. In the results please indicate the number of samples excluded for temperature storage > -196 C.. It looks like 55, but please confirm.

2.     Minor comment in the Dietary fatty acids intake section: Please change “Additionally to dietary intake to “In addition to dietary intake”.

3.     In statistical analysis:

“Plasma fatty acid concentrations were expressed as a percentage of total fatty acids (%TFA) in plasma.”  In the table, strongly prefer that the footnote mentions “total plasma fatty acids”. You can keep the TFA abbreviation, but see later comment. It is very easy for the reader to confuse the two forms of FA.

4.      “EPA and DHA are expressed in 100  mg/day, ALA and LA as g/day and AA as 10 mg/day to get most intuitively interpretable results out according to typical daily intakes.”

I wonder if ‘statistically evaluated and presented as’ would be better than “expressed in”? In Table 1 you used grams to express (describe) all dietary FA.

Suggest that this phrase “get most intuitively interpretable results out” be replaced with “ to obtain intuitively interpretable results”.

5.     Related to Table 1, the intake of AA looks low for Danes. Please see Forsyth et. al. Global Estimates of Dietary Intake of Docosahexaenoic Acid and Arachidonic Acid in Developing and Developed Countries Ann Nutr Metab 2016;68:258–267. For Denmark they report:

            [264.6 mg/day AA (this would be 0.26 grams or 0.07 as a % of total energy)].

            Is there any chance that you reported % total energy (0.08) instead of grams per day?

            Populations could differ within Denmark, and these are thin women, but please check.

6.     Throughout the paper including tables such as Table 2, it is more accurate to refer to ‘difference’ instead of change and ‘greater’ instead of increase. The words ‘change’ and ’increase‘ imply change over time but your study is cross-sectional.

7.     From line 193 (Results) “When expressing the intake variable as %TFA intake instead of absolute intake (weight unit/day), the correlation coefficient increased for DHA, which is also demonstrated by lower p-value in the unadjusted linear regression coefficient.”

            For “%TFA” intake I believe you refer to evaluating each fatty acid as a percentage of total fat         intake but better to be explict (whether you used grams or kjoules it will come out the same).             Because  the label is similar to what you used in the table for (plasma) forms (Table 1), it   becomes confusing somehow.  Prefer to not use  “% TFA”  for these two different kinds of   measures.

8.     For Table 2 and 3 please bold ‘Biomarker” (or bold the whole title) to make it clear that these are circulating (plasma).FA and not dietary. Is the only difference between 2 and 3 that one is stratified by ALA median intake (please put ‘dietary’ next to ALA in the table so the reader knows). Or that one has adjustment by “%TFA intake instead of absolute intake”? There is a sentence under Table 3, but unclear which table it applies to (or if to both).  The title for Table 3 could be shorter.

9.     This  sentence refers to “lower P value” which implies that somewhere there is a ‘higher P value”. Where is the lower P value (Table 3)? Where is the higher P value? (not shown?). Either way is okay but the reader may want to see the lower and higher and will look for them (or if you indicate “not shown” then the reader will not look for them).

“When expressing the intake variable as %TFA intake instead of absolute intake (weight unit/day), the correlation coefficient increased for DHA, which is also demonstrated by lower p-value in the unadjusted linear regression coefficient.

10.  Much of the discussion is interesting and informative. However, this sentence is confusing:

“In the present study, total fat intake was indirectly accounted for by adjusting for total energy intake; furthermore, expressing the intake variable as percentage of total fat intake maintained a correlation between LCn3FAs intakes and their biomarkers with strengthened correlation for DHA”

I am not sure that adjusting for total energy accounts for total fat intake.  Fat is 33% of total energy, Carb is 52% and protein is 15%. 52 + 15 = 67% (two thirds of the intake). You are correct that the energy density of fat is twice that of carb and protein, but total energy includes all three.

11.  In the Discussion, you mention genes related to elongation and desaturation. It would educate the reader to name the FADS gene cluster.

12.   In line 267 there is a reference to ‘less specific body functions”. What do you mean?

13.  In line 290 there is reference to ‘lower affinity of the sources for specific FA’? Do you mean that that FFQ cannot capture them accurately? Or something else?

14.  Please tell the reader the primary oil consumed in this population. Which FAs does it supply?

15.  In place of  “simply natural inconsistency”  (Line 316) suggest “biologically based variability”.

Author Response

Thank you very much for this thorough and very useful review!

1.     The Section on secondary analyses mentions 3 storage temperatures, -20 C, -80 C and -196 C. Please indicate storage temperature in the methods for “Plasma FA concentration” section, where readers will look for it. In the results please indicate the number of samples excluded for temperature storage > -196 C.. It looks like 55, but please confirm.

Agreed and moved to section about Plasma FA concentration in methods. Number of excluded samples are specified.

2.       Minor comment in the Dietary fatty acids intake section: Please change “Additionally to dietary intake to “In addition to dietary intake”.

Agreed and changed.

3.        In statistical analysis: “Plasma fatty acid concentrations were expressed as a percentage of total fatty acids (%TFA) in plasma.”  In the table, strongly prefer that the footnote mentions “total plasma fatty acids”. You can keep the TFA abbreviation, but see later comment. It is very easy for the reader to confuse the two forms of FA.

Agreed and foot note inserted in tables. %TFA when referring to percentage of total plasma fatty acids have been changed to %TPFA. The abbreviation for %TFA when referring to intake has been taken out. See attached manuscript is this is ok.

4.     “EPA and DHA are expressed in 100 mg/day, ALA and LA as g/day and AA as 10 mg/day to get most intuitively interpretable results out according to typical daily intakes.”.

I wonder if ‘statistically evaluated and presented as’ would be better than “expressed in”? In Table 1 you used grams to express (describe) all dietary FA.

Agreed and changed

Suggest that this phrase “get most intuitively interpretable results out” be replaced with “ to obtain intuitively interpretable results”.

Agreed and changed

5.     Related to Table 1, the intake of AA looks low for Danes. Please see Forsyth et. al. Global Estimates of Dietary Intake of Docosahexaenoic Acid and Arachidonic Acid in Developing and Developed Countries Ann Nutr Metab 2016;68:258–267. For Denmark they report:

[264.6 mg/day AA (this would be 0.26 grams or 0.07 as a % of total energy)].

Is there any chance that you reported % total energy (0.08) instead of grams per day?

Populations could differ within Denmark, and these are thin women, but please check.

I’ve checked the intake within the whole DNBC population and this gives similar median intake of arachidonic acid so the estimate seems right in that way. I’m sure that the fact that they are younger women (and pregnant) is a reason to the lower intake of animal products (and they also lower total energy intake) compared to the general Danish population with both men and women.

6.     Throughout the paper including tables such as Table 2, it is more accurate to refer to ‘difference’ instead of change and ‘greater’ instead of increase. The words ‘change’ and ’increase‘ imply change over time but your study is cross-sectional.

Agreed and changed. Have kept the word “increase” in the tables showing plasma concentration in response to increase in daily intake, since I find it more understandable this way. Suggestions for other wording are very welcomed!

7.     From line 193 (Results) “When expressing the intake variable as %TFA intake instead of absolute intake (weight unit/day), the correlation coefficient increased for DHA, which is also demonstrated by lower p-value in the unadjusted linear regression coefficient.”

For “%TFA” intake I believe you refer to evaluating each fatty acid as a percentage of total fat         intake but better to be explict (whether you used grams or kjoules it will come out the same).             Because  the label is similar to what you used in the table for (plasma) forms (Table 1), it   becomes confusing somehow.  Prefer to not use  “% TFA”  for these two different kinds of   measures.

Agreed and changed (see point 3)

8.     For Table 2 and 3 please bold ‘Biomarker” (or bold the whole title) to make it clear that these are circulating (plasma) Ageed and changed. FA and not dietary. Is the only difference between 2 and 3 that one is stratified by ALA median intake (please put ‘dietary’ next to ALA in the table so the reader knows) Agreed and changed (see attached manuscript if its ok). Or that one has adjustment by “%TFA intake instead of absolute intake”? There is a sentence under Table 3, but unclear which table it applies to (or if to both). Clarified. The title for Table 3 could be shorter Agreed and some text has been moved to footer of table instead since it’s crucial for understanding.

9.     This sentence refers to “lower P value” which implies that somewhere there is a ‘higher P value”. Where is the lower P value (Table 3)? Where is the higher P value? (not shown?). Either way is okay but the reader may want to see the lower and higher and will look for them (or if you indicate “not shown” then the reader will not look for them).

“When expressing the intake variable as %TFA intake instead of absolute intake (weight unit/day), the correlation coefficient increased for DHA, which is also demonstrated by lower p-value in the unadjusted linear regression coefficient.

Agreed it can be hard to understand what is meant – clearer table indication inserted.

10.  Much of the discussion is interesting and informative. However, this sentence is confusing:

“In the present study, total fat intake was indirectly accounted for by adjusting for total energy intake; furthermore, expressing the intake variable as percentage of total fat intake maintained a correlation between LCn3FAs intakes and their biomarkers with strengthened correlation for DHA”

I am not sure that adjusting for total energy accounts for total fat intake.  Fat is 33% of total energy, Carb is 52% and protein is 15%. 52 + 15 = 67% (two thirds of the intake). You are correct that the energy density of fat is twice that of carb and protein, but total energy includes all three.

Yes, agreed. The total energy intake is not the best way to adjust for fat intake. This part has been deleted and the argumentation is solely based on secondary analysis using percentage of total fatty acid intake instead of g/day.

11.  In the Discussion, you mention genes related to elongation and desaturation. It would educate the reader to name the FADS gene cluster.

Good idea, it’s added to the text.

12.  In line 267 there is a reference to ‘less specific body functions”. What do you mean?

I’ve revised this sentence – hope it’s more precise now (see attached manuscript).

13.  In line 290 there is reference to ‘lower affinity of the sources for specific FA’? Do you mean that that FFQ cannot capture them accurately? Or something else?

Yes, that is what I meant – I’ve tried to rephrase it, hope it helped (see attached manuscript).

14.  Please tell the reader the primary oil consumed in this population. Which FAs does it supply?

Inserted in the method section “dietary fatty acid intake” section and in the discussion.

15.  In place of  “simply natural inconsistency”  (Line 316) suggest “biologically based variability”.

Agreed and changed.  

Have a good day!

Marie